Age and socio-economic status affect dengue and COVID-19 incidence: spatio-temporal analysis of the 2020 syndemic in Buenos Aires City

Carbajo Aníbal E. 1 2
Cardo María V. 1 2
Pesce Martina 3
Iummato Luciana E. 3
Bárcena Barbeira Pilar 3
Santini María Soledad 2 4
Utgés María Eugenia mutges@anlis.gob.ar 5
1 Instituto de Investigación e Ingeniería Ambiental (IIIA), Escuela de Hábitat y Sostenibilidad, Universidad Nacional de San Martín (UNSAM) , San Martín , Buenos Aires , Argentina
2 Consejo Nacional de Investigaciones Científicas y Técnicas (CONICET) , Buenos Aires , Argentina
3 Dirección Nacional de Epidemiología e Información Estratégica, Ministerio de Salud de la Nación , Buenos Aires , Argentina
4 Instituto Nacional de Parasitología (INP), ANLIS “Dr. C. G. Malbrán”, Ministerio de Salud de la Nación , Buenos Aires , Argentina
5 Centro Nacional de Diagnóstico e Investigación en Endemo-epidemias (CeNDIE), ANLIS “Dr. C.G. Malbrán”, Ministerio de Salud de la Nación , Buenos Aires , Argentina
Nishiura Hiroshi
Electronic publication date: 2023 Sep 22
Publication date: 2023
Volume: 11
Electronic Location ID: e14735
Received 2022 Sep 8; Accepted 2022 Dec 21
Copyright: ©2023 Carbajo et al.
Copyright year: 2023
Copyright holder: Carbajo et al.
License: This is an open access article distributed under the terms of the Creative Commons Attribution License, which permits unrestricted use, distribution, reproduction and adaptation in any medium and for any purpose provided that it is properly attributed. For attribution, the original author(s), title, publication source (PeerJ) and either DOI or URL of the article must be cited.
License URL: https://creativecommons.org/licenses/by/4.0/

Keywords: Syndemic, SARS-CoV2, Arbovirus, Compulsory social isolation, Pandemia, Temperate Argentina, Precarious settlement, Slum, Socio-economic stratum

Funding: The authors received no funding for this work.

==============================
In early 2020, Argentina experienced the worst dengue outbreak in its history, concomitant with first-to-date increasing COVID-19 cases. Dengue epidemics in temperate Argentina have already been described as spatially heterogeneous; in the previous 2016 outbreak, transmission occurred 7.3 times more frequently in slums compared to the rest of Buenos Aires City (CABA). These informal settlements have deficient sanitary conditions, precarious housing and high incidence of social vulnerabilities. The purpose of this work was to study the spatio-temporal patterns of the 2020 dengue epidemic in CABA in relation to socio-economic living conditions of its inhabitants and its interaction with the onset of COVID-19. The study considered the period between Jan 1st and May 30th 2020. Dengue and COVID-19 databases were obtained from the National Health Surveillance System; each record was anonymized and geo-localized. The city was divided according to census tracts and grouped in four socio-economic strata: slums, high, mid and low residential. An aligned-rank transform ANOVA was performed to test for differences in the incidence of dengue and COVID-19, and age at death due to COVID-19, among socio-economic strata, four age categories and their interaction. The incidence by cluster was calculated with a distance matrix up to 600 m from the centroid. Spatial joint dengue and COVID-19 risk was estimated by multiplying the nominal risk for each disease, defined from 1 (low) to 5 (high) according to their quantiles. During the study period, 7,175 dengue cases were registered in CABA (incidence rate 23.3 cases per 10,000 inh), 29.2% of which occurred in slums. During the same period, 8,809 cases of COVID-19 were registered (28.6 cases per 10,000 inh); over half (51.4%) occurred in slums, where the median age of cases (29 years old) was lower than in residential areas (42 years old). The mean age of the deceased was 58 years old in slums compared to 79 years old outside. The percentage of deaths in patients under 60 years old was 56% in slums compared to 8% in the rest of the city. The incidence of both diseases was higher in slums than in residential areas for most age categories. Spatial patterns were heterogeneous: dengue presented higher incidence values in the southern sector of the city and the west, and low values in highly urbanized quarters, whereas COVID-19 presented higher values in the east, south, high populated areas and slums. The lowest joint risk clusters were located mainly in high residential areas, whereas high joint risk was observed mainly in the south, some western clusters, the historical part of the city and center north. The social epidemiological perspective of dengue and COVID-19 differed, given that socio environmental heterogeneity influenced the burden of both viruses in a different manner. Despite the overwhelming effect of the COVID-19 pandemic, health care towards other diseases, especially in territories with pre-existing vulnerabilities, should not be unattended.

Introduction

SARS-CoV2 was detected and notified on December 31st, 2019, generating a new infection named COVID-19. This virus belongs to the Coronavirus family known for causing upper-respiratory tract illnesses (NIH, 2021). First cases were detected at Wuhan, China, the epicenter of the outbreak, and then extended to other countries in a quick propagation at community, regional and international scale causing an exponential increase in cases and deaths (OPS, 2020a).

The first case in Argentina was reported on March 3rd, 2020 at Buenos Aires Autonomous City (CABA, for its acronym in Spanish) (MSN, 2020a). At the beginning, cases were notified in people returning to the country from areas with viral circulation and their close contacts but, by the end of the month, community circulation was registered both in the metropolitan area of Buenos Aires and in the north-east of the country. Schools were shut down on March 16th together with international airports and land border closure for foreigners. On March 20th, the entire country population with the exception of essential workers was isolated at home, and border closure extended to all Argentinian residents (this group of measures was locally denominated ASPO). The majority of the population adhered to these measures, so that public transport journeys within CABA decreased 65% for buses and trains, and 94% for subways; the entry/exit of private vehicles to the city fell 85% and internal circulation was reduced by 68% (GCBA, 2020a). During early April, first cases in CABA slums were reported to the National System of Health Surveillance. In May, the National Health Ministry in collaboration with the City Health Ministry established a testing strategy in slums to contain the situation, denominated DETeCTAr. It consisted in a home-by-home active search for close contacts of confirmed cases and all citizens with COVID-19 compatible symptoms, followed by the redirection to a nearby testing site for PCR diagnostic (MSN, 2020b). ASPO was maintained until the beginning of June, after which isolating measures were progressively relaxed allowing certain types of commercial and recreational activities (MNS, 2020c).

Taking into consideration the regional epidemiology of arboviral diseases, multiple implications arose concomitant with COVID-19. In 2020 Argentina experienced the worst dengue outbreak in its history, which largely exceeded previous epidemics of 2009 and 2016. The epidemic took place between January and May, similar to the two mentioned outbreaks, with co-circulation of three serotypes (72% DEN-1, 26% DEN-4 y 2% DEN-2) (MSN, 2020d). In non-endemic territories like Argentina, dengue transmission dynamics is influenced mainly by the density of its vector, Aedes aegypti, by climatic factors that condition its ecophysiology, and by people’s movements from endemic to non-endemic areas. Typically, outbreaks start in towns bordering endemic neighboring countries (Bolivia, Paraguay and Brazil) and then extend to temperate regions (Carbajo, Cardo & Vezzani, 2012; Carbajo et al., 2018a). In the Buenos Aires Metropolitan Area, imported cases have been positively correlated with a higher proportion of foreign population and larger working-age class, confirming virus pressure from neighboring countries related to travelers who usually live in low-income neighborhoods (Carbajo et al., 2018b). In the 2016 outbreak, transmission occurred 7.3 times more frequently in slums compared to the rest of CABA, and most foci were more persistent within slums than outside them (Gurevitz et al., 2021). This may be explained by the fact that in slums, irregular water supply, precarious toilets in the houses and lack of regular garbage collection foster the proliferation of potential larval habitats for Ae. aegypti, mainly reusable buckets, tanks for water reservoirs and disposable containers which accumulate rain water (Gubler, 2011).

As dengue and COVID-19 share clinical symptoms (e.g., fever, myalgia, fatigue, headache, rash), the differential diagnosis in a situation of syndemic is not straightforward (Harapan et al., 2021, but see Thein et al., 2021). Co-infections have been reported in individual patients -both in Argentina (Radisic et al., 2020; Salvo et al., 2020) and abroad (e.g., Epelboin et al., 2020; Verduyn et al., 2020; Villamil-Gómez et al., 2021)- and in small cohorts (e.g., Carosella et al., 2021; Mejía-Parra et al., 2021). As both diseases may potentially lead to fatal outcomes especially in patients with chronic comorbidities, overlapping infections may increase the number of patients requiring intensive care and mechanical ventilation, further stressing the health attention system (Cardona-Ospina et al., 2020).

Three million people live in CABA, which together with the neighboring contiguous urbanization adds up 13 million inhabitants, making it the largest urban area in the country and the second in South America. The socio-spatial dynamics of its historic conformation has generated boundaries that define contrasting living conditions for its inhabitants, showing a wide gap between the population living in precarious settlements and the one from the rest of the city. Dadamia (2019) estimated that 170,000 people live in 30 slums occupying an approximate area of 5 km2. These territories are generally poorly integrated to the urban plot and basic service networks, resulting in deficient sanitary conditions and precarious housing which, added to poor nutrition quality, low health coverage and limited economic possibilities to face contingencies, reinforce the social vulnerability of its inhabitants. A greater exposure to threats combined with higher levels of vulnerability therefore results in a greater environmental risk in slums than the one encountered in a formal neighborhood (DPNA, 2018; Morandeira et al., 2019). Slums therefore deserve special attention due to unequal conditions to face diseases, favoring both infections and severe illness, e.g., structural poverty, high prevalence of comorbidities, and impossibility of maintaining social distancing and recommended hygiene measures. The need for a singular approach to the COVID-19 pandemia in these contexts has been globally alerted (Corburn et al., 2020; Austrian et al., 2020; Naciones Unidas-Argentina, 2020).

The purpose of this work was to study the spatio-temporal patterns of the 2020 dengue epidemic in CABA in relation to socio-economic living conditions of its inhabitants and its interaction with the onset of COVID-19. The specific objectives were (1) to compare the spatial pattern of COVID-19 and dengue outbreaks, considering possible differences in the epidemiological profiles in and outside slums; and (2) to evaluate the impact of preventive and compulsory social isolation implemented for COVID-19 on the temporal patterns of dengue cases by socio-economic stratum. The associated hypotheses were that (1) the total and per class age incidence of COVID-19 and dengue differ by socio-economic stratum; and (2) the establishment of ASPO to contain COVID-19 decreased the incidence of dengue cases. The unprecedented national isolation and border closure due to COVID-19 spread in the midst of the dengue epidemic can provide relevant information regarding dengue transmission dynamics and possible keys to its control and mitigation.

Materials and Methods

Study area and timespan

Buenos Aires Autonomous City (CABA) is the capital city of Argentina, located in the center-east of the country with an area of 203 km2. The urban plot resembles a hand fan of 60 km of perimeter that limits to the south, west and north with urbanized areas of Buenos Aires Province and to the east with the Río de la Plata river (Fig. 1). It is formed by 48 neighborhoods and 15 administrative communes. For operational purposes, it is divided in 3554 census tracts (defined as the minimal sampling unit for the last national census performed in 2010, hereafter CT), each consisting of a polygon of variable size comprising in average 400 dwellings (min. 6, max. 1405).

Figure 1 Classification of census tracts in four socioeconomic strata.

(A) Slums (numbered from 1 to 17 for identification purposes) and (B) three residentials. Parks and other vegetated areas, and water bodies were excluded from the classification and shown in gray. Slum numbers: s1 Villa 1-11-14/Rivadavia; s2 Villa 21-24/Zavaleta; s3 Villa 31/31bis/ San Martín; s4 Fátima/Carrillo; s5 Villa 20; s6 Ciudad oculta; s7 Cildañez; s8 Barrio INTA/María Auxiliadora/Barrio Obrero/Bermejo; s9 Pirelli; s10 Playón de Chacarita/Villa Fraga; s11 Barrio Rodrigo Bueno; s12 Lamadrid; s13 Asentamiento Villa Ortúzar; s14 La Carbonilla; s15 Villa 26; s16 NN2; s17 NN3. Base map ©2022 Google, Data SIO, NOAA, U. S. Navy, NGA, GEBCO, Maxar Technologies, Landsat/Copernicus, Terra Metrics.

The last national census recorded 2,890,151 inhabitants with 6.3% of residents in slums. Mean population per CT was 814 inh (min. 25, max. 3945) (INDEC, 2010). Projections for 2020 (total population 3,075,646 inh) estimate that CABA has 11% more women than men, and that the population over 60 years old (hereafter, yo) represents over 21% of the total, while those under 15 yo comprise almost 20% (GCBA, 2020b). The population living in slums has a younger profile (43% < 15 yo, and 8% > 60 yo; Bonfiglio et al., 2017). The global average age of the population is 42 and 37 yo, while life expectancy is 82 and 75 yo for women and men, respectively. In slums, the average age is lower (e.g., in 2009 the average age in the slum known as “Villa 31” was 23.3 yo) (DGEyC, 2009).

Slum inhabitants are unevenly distributed within CABA, 88% of them are located in the southern part of the city, and surrounded by low-income neighborhoods that do not fit the slum definition. The wealthier population is located mainly in a fringe from east to north, some areas of the north-west and the geographic center of the city (Fig. 1). The spatial distribution of socio-economic strata has been previously studied in detail (e.g., Marcos, Mera & Di Virgilio, 2015).

The current study considered the period between January 1st and May 30th 2020, that is, between epidemiological weeks (EW) 1 and 22. Thus, the entire dengue epidemic was analyzed from the beginning to the drop in the number of cases simultaneously with the onset of the COVID-19 epidemic. During the study period, DETeCTAr was established in slums s3, s1 and s2 (as numbered in Fig. 1A, base map data ©2022 Google) on May 5th, 11th and 18th respectively.

Databases

Databases were obtained from the National Health Surveillance System, which is the legitimate source of epidemiological information at the national level as it collects all obligatory notification health events reported by enabled users from the public and private sectors and the social security system (MSN, 2022). Each record (a health event or case) was anonymized with a numerical identifier and geo-localized as described ahead. The variables studied for each case were: age (continuous for general descriptions and in four categories for further analyses: 0–19, 20–39, 40–59, and over 59 yo), sex (male/female), and death (yes/no).

Dengue database

All cases with a symptom onset date between January 1st and May 30th, 2020 were considered. If not recorded, the earliest date among consultation, sample collection for diagnosis and case notification date was used. A suspicious case is considered as a person with fever, lasting less than 7 days without upper airway involvement or other defined etiology, accompanied by two or more of the following signs: headache and/or retroocular pain, general malaise, myoarthralgia, diarrhea, vomiting, anorexia and nausea, skin rash, petechiae or a positive tourniquet test, leukopenia, thrombocytopenia. A probable case is any person with a positive NS1 antigen detection test or a positive IgM detection test, whereas a confirmed case is any suspicious case with a confirmatory laboratory diagnosis (viral isolation/genome detection/neutralization test) or an epidemiological link, depending on the situation of the jurisdiction (MSN, 2015).

Regarding the origin, an imported case is any confirmed or probable case with address in an area without viral circulation and that has visited areas with proven viral circulation in the last 15 days, whereas an autochthonous case is such whose possible site of infection corresponds to the jurisdiction of habitual residence of the patient.

All probable and confirmed cases as defined above were included in the study, both autochthonous and imported, and pooled for further analyses.

COVID-19 database

Cases with a symptom onset date between March 1st and May 30th 2020 were selected. If not recorded, the earliest date of sampling for diagnosis was considered or, failing that, the date of notification of the case.

A suspicious case is defined as any person with fever of at least 37.5 °C together with one or more of the following symptoms: cough, sore throat, respiratory distress or alteration of taste or smell. For closed institutions, close contact in slums and health personnel, only 1 of the specified symptoms is required (MSN, 2020e). A confirmed case is any suspicious case confirmed by PCR laboratory test (MSN, 2020e). These case definitions correspond to the study period and have been modified since then.

Case geo-reference

A slum was operationally defined as a neighborhood in which at least eight grouped or contiguous families live, where more than half of the population does not have land title and/or lack regular access to at least two of the three basic public services (running water network, electricity network with household meter, and sewer network) (InfoLeg, 2017). Combining the slums shapefile and the map of vulnerable neighborhoods of CABA (GCBA, 2015), all CTs were assigned to 2 categories, slums and non-slums. Taking a conservative criterion, if any CT contained a portion of slum it was classified as such. Next, contiguous slum CTs were put together into a single cluster, giving a total of 17 slums.

In the National Health Surveillance System databases, the address is a free field. In slums, this field generally specified the slum’s name, the block’s number and the house’s number. This allowed us to identify that a case occurred in a slum, but not to geolocate it accurately. A list of words and fragments of words indicating slums was created to identify them. A macro was developed in Visual Basic to scan the address field for the presence of any of the terms in the list and thus assign the cases to slums. As a result, all cases were classified into: 1- located in one of the 17 identified slums, 2- located in unspecified slums (without the slum’s name) or 3- outside slums, when no term on the list was found in the address field. The cases in 2- were manually inspected and, if possible, assigned to their corresponding slum based on knowledge of the study area. All cases in 3- were georeferenced in R version 4.1.0 (R Core Team, 2021) using the RUMBA package (v0.1.0, Vazquez Brust, 2019), converting addresses in lat/lon variables. Finally, a union of the cases was made with the map of CTs with the slums information. In this way, the CT was assigned to the records and, if they were within slum CTs, they were included within that classification.

Classification of non-slums CTs in three socio-economic strata

Non-slums CTs were classified in 3 socio-economic strata using available cartography for the study area based on 4 variables, as follows: proportion of households with water from the public network inside the home; proportion of households that use the gas network, bulk gas, or gas tube primarily as cooking fuel; proportion of households with a computer; and population from 25 to 64 yo with complete university education (Marcos, Mera & Di Virgilio, 2015). The original categories High Residential (HighR), Mid Residential (MidR) and Low Residential (LowR) were kept as defined in the cited work as three levels in the socio-economic gradient. To reclassify the CTs that originally belonged to categories as Central City, Colonial City and Housing Complexes, a decision tree classifier was used with the 4 explanatory variables mentioned above and the response variable as one of HighR, MidR, LowR, or slum + informal settlement + transitory housing complex (as in Marcos, Mera & Di Virgilio, 2015). The cutoff criteria obtained for that subset were applied to the above to reclassify them. Not urbanized areas (parks and water bodies) were excluded from the classification.

Most (67/83) of the CTs from Central City (located at the north-east edge of the study area) were assigned to MidR, while two-thirds (57 out of 86) of the CTs from Colonial City (also located east, south of Central City) were classified as HighR. The housing complexes (located mostly in the southern part of the city but also on the west bank) were distributed equally between LowR and MidR (Table 1). The total number of CTs for each residential category was 401, 1559 and 1439 for LowR, MidR and HighR respectively, resulting in the final classification of four socio-economic strata observed in Fig. 1B (census tracts geospatial data from INDEC, 0000).

Table 1 Categorization of census tracts’ original classes by Marcos, Mera & Di Virgilio (2015) into three socio-economic residential levels (HighR, MidR, LowR) and slums.

		Initial classification			
Final category		Colonial city	Central city	Housing complex		Total	
HighR		57	10	7		74	
MidR		27	67	49		143	
LowR		2	6	54		62	
Slum				8		8	
Total		86	83	118		287	

Non spatial analysis

Socio-economic strata x age

Aligned-Rank Transform ANOVA was performed using packages ARTool (Kay et al., 2021) and rcompanion (Mangiafico, 2016) in R to test for differences in the incidence of dengue and COVID-19, and age at death due to COVID-19 among socio-economic strata, age categories and their interaction. Post-hoc comparisons for pairwise interactions were conducted, with Bonferroni adjustment for multiple pairs.

Temporal dengue pattern

The proportion of dengue cases in each EW were plotted together with weekly mean temperature to compare the occurrence of dengue cases by socioeconomic strata. The incidence of dengue cases per EW by 10,000 inhabitants was also plotted to help visualize any effect due to the ASPO and COVID-19 disturbances.

Spatial analysis

Variogram

We calculated the sample variogram of the number of dengue cases per 10,000 inh during the study period in order to have a picture of the spatial dependence of dengue cases per CT as a function of distance. We used an isotropic Cressie’s robust variogram estimate included in the R gstat package (Pebesma, 2004). Three parameters are important in the interpretation of these graphs: the nugget, the sill and the range. The nugget is the height of the jump of the variogram at the origin, which represents basal variability at very close separation distances. The sill is the variability due to spatial correlation, and is the difference between total variation and the nugget. The range represents the distance limit beyond which the data are no longer spatially correlated and is calculated as the distance where the variance first reaches its maximum (Cressie, 1993).

Incidence by cluster

A distance matrix was run in QGIS 3.16 from the centroid of each CT to all its neighboring CTs up to 600 m. The selection of this distance was to make clusters similar in extension to CT agglomerates that make up the largest slums, and also following the report that this was approximately the correlation distance for dengue cases in CABA during the 2016 epidemic (Gurevitz et al., 2021). This was also intended to make population totals in and outside slums within the same magnitude order and therefore be able to compare incidence values in the different socio-economic strata.

Due to the exclusion of non-inhabited areas, there are 5 CT (3 classified as slum and 2 as LowR) that have no neighbors within 600 m, for which the value of the CT itself was applied. For all others, the cumulative population and cumulative cases of dengue and COVID-19 were calculated for each cluster. With this information the incidence was calculated as:

Incidence_clusteri = accumulated cases_clusteri * 10,000/accumulated population_clusteri

For both diseases, each cluster was assigned to one of 5 categories according to the quantiles of its incidence throughout the whole study period (EW 1–22). Categories were ranked nominally from 1 (lowest) to 5 (highest) incidences of each disease. To study the combined behavior of both diseases in space, nominal categories for each disease were multiplied (COVID-19 category * dengue category) obtaining values from 1 (lowest) to 25 (highest) combined risk. We considered values 1 and 2 indicative of low joint risk and values 20 and 25 of high risk.

The incidence by residential type (HighR, MidR, LowR) was compared with the one registered in slums, evaluating whether the values in slums were higher than the 97.5% quartile of the distribution of values obtained in the residential clusters. First, the analysis was performed without taking into account the composition of CTs of each cluster, then considering whether the neighbors included in each cluster were of the same residential type as the centroid CT.

Results

Between January 1st and May 30th 2020, 7,175 dengue cases were registered in CABA, the incidence rate being 23.3 cases per 10,000 inhabitants. No deaths were recorded and only 172 cases showed signs of alarm. The database was fairly complete, allowing the georeference of 93% of the cases, and lacking information of gender and age only for 3 and 9 cases, respectively. Regarding gender, 50.2% of the cases were men and 49.8% were women, and the median age of all cases was 33 yo (32 yo men; 34 yo women). Of the total geo-referenced cases (6,670), 29.2% occurred in slums. For the former, the median age was 29 yo and the percentage of women was 52.5% while in the rest of the city values were 35 yo and 48.2%. The largest number of cases in slums occurred in s1, s2 and s5 (Fig. 1), accumulating among them 67% of the slum cases.

During the same period, 8,809 cases of COVID-19 were registered, with an incidence rate of 28.6 cases per 10,000 inhabitants. There were 251 deceased persons, with a mortality rate of 0.82 per 10,000 inhabitants and a fatality rate of 2.8%. Georeference was achieved for 87% of the cases, all records reported age information and 38 lacked gender information. Cases were distributed almost equally between men and women (49.6% and 50.4%), the median age of the cases was 34 yo (34 yo women; 35 yo men). As for the deceased, 55.1% were males (median age 70 yo) and 44.9% were females (83 yo). Of the total georeferenced cases (7,653), slightly over half (51.4%) occurred in slums. There were almost no variations in and outside slums regarding sex distribution, but the median age of cases was lower in slums (29 yo) than outside (42 yo). Greater differences were observed among the deceased, the proportion of deaths among women in slums was 32% (14/44) compared to 49% outside slums (83/171).

Age groups, cases and mortality

Dengue

In the analysis of dengue incidence, there was a significant interaction between the socio-economic stratum and the age group (F = 37.28, p < 0.001). In general, the incidence was higher in slums for all age categories, in LowR for all categories below 60 yo, and in MidR for 20–59 yo. In MidR and HighR, intermediate age categories presented higher incidences than the age extremes. In HighR, the youngest and the eldest had remarkably low incidence, whereas incidence for intermediate age categories was comparable to >59 yo in LowR and MidR (Table 2).

Table 2 Dengue and COVID-19 incidence per age group and socio-economic stratum, along with age statistics of diseased due to COVID-19.

yo stands for years-old. Within a column, different superscript letters indicate significant differences (p < 0.05) in pairwise comparisons.

			COVID-19	
Socio-economic stratum	Age group	Dengue Incidence (mean ± SE)	Incidence (mean ± SE)	Diseased age (mean ± SE)	Diseased %<60 yo (total all ages)	
Slums				58.1 ± 13.4a	56.3 (48)	
	0–19 yo	37.4 ± 11.4a,b	66.9 ± 29.5a,b			
	20–39 yo	66.4 ± 18.4a	106.3 ± 46.7a			
	40–59 yo	65.6 ± 18.6a,b	132.2 ± 63.8a			
	>59 yo	73.8 ± 27.4a,b	151.6 ± 93.7a,b,c,d,e			
LowR				66.7 ± 16.8a	35.3 (17)	
	0–19 yo	23.8 ± 2.4b	12.6 ± 2.0g,h			
	20–39 yo	40.7 ± 3.5a	29.4 ± 2.5a,b			
	40–59 yo	36.8 ± 3.2a,b	29.1 ± 3.3b,e			
	>59 yo	13.3 ± 1.8d,e	18.4 ± 3.4c,g			
MidR				80.9 ± 13.6b	8.7 (92)	
	0–19 yo	18.6 ± 1.4d	6.6 ± 0.7f,h			
	20–39 yo	26.1 ± 1.4b	16.3 ± 0.9d,e			
	40–59 yo	27.9 ± 1.6b	17.3 ± 1.2c,d			
	>59 yo	10.7 ± 1.0e	15.9 ± 1.9g			
HighR				81.4 ± 9.7b	1.5 (68)	
	0–19 yo	5.8 ± 0.7c	3.7 ± 0.7f			
	20–39 yo	10.4 ± 0.7d	11.9 ± 0.7c,d			
	40–59 yo	11.5 ± 0.9d,e	13.8 ± 0.9c,g			
	>59 yo	4.5 ± 0.5c	13.9 ± 1.5c,g			

COVID-19

There was also a significant interaction between the socio-economic stratum and the age group (F = 53.68, p < 0.001). Incidence was remarkably higher in slums than in residential CTs for all age groups, however large intra-class variability prevailed in the >59 yo group (Table 2). Within each residential type, incidence for the 0–19 yo group was remarkably lower than for the 20+ yo groups, whereas incidence for mid-age groups in LowR was higher than for all MidR and HighR (with the exception of MidR 20-39 yo, Table 2). Comparing age groups among socio-economic strata, incidence in the group 0-19 yo was maximum for slums and LowR, while in MidR and HighR showed the lowest values. The elder population (>59 yo) incidence was lowest in slums, intermediate in LowR and MidR and highest in HighR. Of the 251 diseased, 48 were reported in slums, 17 in LowR, 92 in MidR and 68 in HighR, and 26 could not be assigned to a location (and were further excluded). The mean age of the diseased differed significantly among socio-economic strata (F = 37.19, p < 0.001), being higher in HighR and MidR than in LowR and slums. This is also reflected in the proportion of diseased below 60 yo, which was over 56% in slums and less than 2% in HighR (Table 2).

Temporal patterns

Dengue

The general pattern of dengue occurrence showed two main peaks in EWs 11 and 14, and a third in EW 18 for MidR and LowR, and a valley in EWs 12 and 13 which coincided with the establishment of ASPO. The fall in incidence from EW 11 to EWs 12 and 13 was higher in slums as compared to the other strata. All four categories reached the highest peak in EW 14, after which cases in slums decayed until the end of the study, but in all residential categories a rebound in EW 18 was observed (Figs. 2A, 2B). Peaks in EWs 11, 14 and 18 were lagged one week with relative rises in mean temperature. The fall in temperature in EWs 11, 12 and 14 may have been related to the fall in cases the following weeks. But LowR and MidR showed a rise in the proportion of cases in EW12. Also, although the fall in mean temperature to 23 °C in EW 11 was followed by a fall in cases, the fall in EW 7 was followed by a rise. Furthermore, the rise in temperature of EW 13 was 23 °C too (Figs. 2A, 2B).

Figure 2 Temporal dynamics of dengue and COVID-19 from epidemiological week 1 to 22, 2020, in Buenos Aires Autonomous City, Argentina.

Lines show the weekly proportion of total dengue (A) and COVID-19 (C) cases, and the weekly incidence (cases per 10,000 inhabitants) for dengue (B) and COVID-19 (D) by socio-economic stratum, superimposed on bars representing weekly mean temperature.

COVID-19

First cases appeared earlier (EW 10) in residential areas than in slums (EW 13). Then on until the end of the study, cases in slums increased steadily, whereas an oscillation pattern with a rising tendency was observed in residentials (Fig. 2C). The incidence temporal pattern showed the preponderance of cases within slums, with a sharp rise as from EW 17 (Fig. 2D). Note that weekly incidence values for COVID-19 were nearly 4 times higher than for dengue (Figs. 2B, 2D).

Variograms

The range of the variogram for the number of dengue cases per 10,000 inhabitants during the study period was 0.00497°, indicating spatial correlation up to ≈ 550 m (Fig. 3). The nugget was 119.0, and the sill 75.6. The former value suggests a variation of dengue cases between closest CTs around 58%, and the latter that similarity due to proximity accounts for 37% of the variation in dengue cases between CTs (76/(119+76)), which in turn only happens up to 550 m.

Figure 3 Semivariogram of the number of dengue cases per 10,000 inhabitants by census tract (pooled from EW 1 to 22, 2020).

Incidence per cluster

The mean accumulated population per cluster was 10,773 inh (min 105; max 48,427) for residential central CTs and 18,955 (min 4,284; max 41,354) for slums. Regarding the composition of neighboring CTs within each cluster, only 47% (187/399) of the LowR stratum clusters (as defined by their central CT) had over 50% of their neighboring CTs of the same category. Contrastingly, the same stratum neighbors were 82% for MidR and 86% for HighR clusters (1,278/1,559 and 1,239/1,439 respectively). Therefore, for LowR analyses were discriminated regarding neighbor composition, whereas for MidR and HighR it was performed all together.

Dengue

Cluster incidence presented a heterogeneous pattern, with higher values in the southern sector of the city, followed by the west, and low values in highly urbanized neighborhoods (Fig. 4A). Median values were 24.2 for slums, 14.1 for LowR (8.2, 14.2 and 26.2 if discriminated by neighbor composition, less than 0.4, 0.4−0.6 and over 0.6 respectively), 11.6 for MidR and 4.3 for HighR. Slums with higher incidence were s1 (158.1), s5 (137.8), s2 (111.4) and s4 (59.1). Quantiles 2.5–97.5% for each socio-economic stratum were 2.1–118.8 for LowR, 2.2–112.1 for MidR, and 8.8–49.2 for HighR. Only 2 of the 17 slums (s1 and s5) presented values above the 97.5% quantile for LowR and MidR, whereas 4 of them did so for HighR.

Figure 4 Incidence per cluster (number of cases per 10,000 inhabitants).

Dengue (A) and COVID-19 (B). Cluster value is represented by its central census tract.

COVID-19

Cluster incidence presented a marked geographical pattern, with higher values in the east, south, high populated areas and slums (Fig. 4B). Median values were 21.8 for slums, 19.8 for LowR (19.2 for less than 0.4 LowR neighboring CTs, and 21.2 for 0.4–0.6 and over 0.6), 12.0 for MidR and 10.7 for HighR. Slums with higher incidence values were s3 (685), s1 (332), s2 (98.4) and s5 (42.7). Also, 1 CT from LowR and 1 from MidR which are located next to s1 included the slum as a neighbor and therefore presented incidence values close to 300. Quantiles 2.5–97.5% for each socio-economic stratum were 3.1–51.2 for LowR, 2.0–36.4 for MidR, and 4.5–27.1 for HighR.

Joint risk for dengue and COVID-19

The lowest joint risk clusters were located mainly in HighR areas, whereas high joint risk was observed mainly in the south, the historical/oldest part of the city and center north (Fig. 5). As shown in Fig. 6, the proportion of CTs of each joint risk category by type of neighborhood evidenced a clear pattern of higher risk in LowR and slums, and of lower risk in HighR. The two lowest risk categories (1 and 2) represented less than 7% of slums, LowR and MidR tracts and 19% of HighR tracts. More than 51% of HighR CTs did not reach risk level 4 of either illness (joint risk values of 1, 2, 3, 6 and 9). On the other extreme, the 20 and 25 risk categories represented 65% of slums, 38% of LowR, 12% of MidR and 3% of HighR, with 88% of the slums presenting values of 4 or more in one of the illnesses. The three slums with lower joint risk categories were s11, s9 and s13 (all with nominal risk 1 for COVID-19 and 1, 3 and 4 for dengue respectively).

Figure 5 Joint risk for dengue and COVID-19.

Each census tract is classified from 1 to 25 as a product of dengue and COVID-19 risk (each defined in 5 categories from 1 lowest risk to 5 highest risk, equal quantiles).

Figure 6 Joint risk category of dengue and COVID-19.

Proportion of census tracts classified from 1 to 25 in a joint risk category of dengue and COVID-19 (each defined in 5 categories from 1 lowest risk to 5 highest risk, equal quantiles), for each socio-economic stratum.

Discussion

Cities are socio-environmental mosaics of interconnected neighborhoods with varying local realities, promoting particular conditions for the onset and spread of different diseases (Santos et al., 2020). In CABA, both dengue and COVID-19 epidemics during early 2020 were spatially heterogeneous. Differences were encountered in the incidence of both diseases between slums and residential areas and also among the different slums, indicating that informal settlements do not behave as a unit, rather each one presents its particularities. In fact, the community-based testing strategy was established on different dates according to the cases’ peak in each slum.

The selection of the study period was intended to include the entire dengue epidemic, superimposed with the onset of COVID-19. Temporal dengue patterns at temperate latitudes are typically Gaussian, due mainly to climatic conditions and their effect on the population of the vector Ae. aegypti. The temporal span of the 2020 epidemic resembled previous outbreaks (MSN, 2020d). Although the denominated first COVID-19 wave extended until November 2020, the onset is particularly valuable to analyze a potentially heterogeneous pattern and significant socio-environmental associations that were later overruled by the great number of cases and high contagion rates. Also, national records were more accurate at the beginning of the epidemics than when the healthcare system was saturated.

During the so far worst dengue outbreak in CABA, dengue incidence was higher in low income areas and slums than in mid and high residential neighborhoods. Almost one out of three cases were reported in slums (which occupy less than 3% of the territory), similar to the 2016 epidemic (Gurevitz et al., 2021). The general spatial pattern of cases was also similar to the reported therein, with house prevalent neighborhoods showing favored transmission over high rise buildings areas. It is noteworthy that areas of lower height edifications coincide with areas where higher vector activity has been detected (Carbajo, Curto & Schweigmann, 2006) and do not match the most populated areas, which are closer to the east and downtown. This may be related to the fact that highly impervious surfaces and tall edifications do not favor the mobility of Ae. aegypti and do not offer adequate habitat for immatures or adults, whereas low houses with green gardens or yards allow the vector to enter the inner sector of the blocks, and present better conditions for mosquito proliferation (in terms of shade, humidity, artificial container abundance) (Carbajo, Curto & Schweigmann, 2006). The onset of cases also occurred in the lower socioeconomic strata, in line with previous reports of more imported cases in low income neighborhoods with a higher proportion of foreign population and people at an active age (Carbajo et al., 2018b). As Ae. aegypti is a daytime biter, it is common that people acquire dengue at their work location rather than at home. Although during the dengue epidemic of 2020 compulsory isolation prevented most of the population from going to work, this event occurred after EW 12 when dengue had already spread throughout the city. Therefore, virus transportation between work and home may have occurred during early transmission.

On the other hand, COVID-19 cases appeared first in high-income neighborhoods, and after a time lag of around 3 weeks they disseminated to slums. This may be related to the origin of imported cases from the Northern Hemisphere, and that travel to those far destinations occurred most probably in high and mid income strata. It is worth mentioning that during the most restrictive circulation period, essential workers were excepted from mobility restriction measures. Within this group were people in charge of school and community meal programs, a frequent activity in city slums, along with health workers and delivery personnel. Moreover, as in many other Latin American cities, in CABA there is an upper circuit of the urban economy, based on high technology and information flow, and a “lower circuit” that depends on intensive, informal labor (Santos et al., 2020) that did not stop functioning during the isolation measures. A great proportion of workers in slums are part of the informal sector and were not reached by the official release from work and economic support benefits, therefore had to continue performing their activities.

When putting age in focus, dengue incidence did not differ by age category in slums, while it was higher in middle age categories for residential neighborhoods. Similar results were observed for COVID-19, with no differences in incidence per age category for slums, and higher incidence in the other strata for middle age categories. The difference was that, in residentials, when comparing extreme age categories, the young presented higher dengue incidences than the old, and for COVID-19 the pattern was reversed. Deceased people due to COVID-19 were older in HighR and MidR than in LowR and slums, probably related with a heterogeneous distribution of the older population among neighborhoods and with a different probability of contagion. Although older people had more risk of suffering complications, the feasibility of isolation from the rest of the family given an infection might have been more frequent in populations with more economic resources. Physical distancing is virtually impossible in the overcrowding housing situation of slums. Also, lower income urban residents are more likely to use public transportation, where exposure to the virus is higher than in private vehicles. Moreover, the informal working situation mentioned above forced many low income residents to expose themselves instead of staying at home. Once infected, lower age at death may be related to pre-existing conditions, given that under-resourced populations frequently have less access to high quality health care, they are prone to suffer from more baseline illnesses that are associated with higher COVID-19 mortality, such as diabetes, heart disease, pulmonary affections and cancer (Finch & Hernández Finch, 2020). These non-communicable diseases which were once considered more prevalent in wealthier populations are now major drivers of poverty worldwide, related to low quality diets, precarious houses, and living in highly contaminated areas (Niessen et al., 2018). The lack of access to healthcare facilities may have also led to the death of young adults who could have survived if properly treated. Regarding the difference in mortality by gender between slums and residential areas, a higher mortality of men with respect to women has been documented (reviewed in Bienvenu et al., 2020), which could add to the effect of socio-environmental vulnerabilities on mortality rates (as reported in Ribeiro et al., 2021).

Of course, it is impossible to know what dengue incidence would have been if social isolation had not been imposed, or whether it would have risen farther than the actual recorded values. But what can be stated is that mobility restriction did not produce a decrease in dengue incidence as hypothesized, on the contrary, values continued rising in all socio-economic neighborhood types. At a first glance, the ASPO seems to have influenced transmission, this might have been due to less contagion derived from low mobility or to a diminution in the number of testing. But according to the rise in EW 14, it can be inferred that the decay in dengue cases in EWs 12–13 due to ASPO was in the detection or report of cases but not in the transmission. As already mentioned, the most vulnerable populations depend on the public health system and have no possibilities of virtual assistance or medical follow-up. But this pattern was observed for all socio-economic strata, indicating that the fear of contagion and the compliance of the isolation national decree resulted in a lack of medical consultation and the sub-diagnosis caused by symptom overlapping of both diseases.

At the same time, the relationship of weekly dengue cases with temperature should not be overlooked, recalling that in CABA epidemic patterns are typically Gaussian. There is a fall in mean temperature from EW 10 to 11 and to 12, which rises in EW 13 and falls again thereon. The pattern in cases lags one week with temperature, especially in slums and HighR. The first fall from EW10 to 11, to around 23 °C, may not be related to limiting temperature, as lower temperatures were observed in previous weeks while transmission was rising (EW 7 to 9). On the other hand, from EW 11 to 12, and from EW 13 to 14, temperature falls to around 21 °C. Previous studies on temperature dependent R0 models have shown that low temperature transmission limits may be around 17 to 21° (Mordecai et al., 2017; Mordecai et al., 2019). We believe the first fall in cases after EW 11 is related to the ASPO but not due to a true decrease in transmission, consequence of the factors explained above (great pressure exerted by COVID-19 epidemic on the health attention and management systems, the fear of the population to attend health centers during the pandemia, and the sub-diagnosis caused by symptom overlapping of both diseases). Nevertheless, the following falls in case numbers may be due to the decrease in temperature, particularly after EW 15, when temperatures fall to 17 °C and below.

When analyzing joint dengue and COVID-19 risk, the lowest risk categories were scarce in slums and LowR, whereas the highest categories represented more than half of all slums and almost 40% of LowR. In all, it is evident that although COVID-19 started in HighR due to imported cases, when considering the entire study period both epidemics undoubtedly struck harder in slums and low income neighborhoods. This is in line with the aforementioned concept that greater exposure to threats combined with higher levels of vulnerability results in higher environmental risks (DPNA, 2018; Morandeira et al., 2019). Despite the fact that dengue mortality rates in temperate Argentina are remarkably low, in general early clinical detection and adequate treatment of patients can significantly reduce acute cases. Therefore, the combined impact of both epidemics could harm specially the most vulnerable populations (Mascarenhas et al., 2020). Still, the effect of co-infection of dengue and COVID-19 on human health is poorly understood (but see Carosella et al., 2021). The first-time occurrence of a high magnitude syndemic in Buenos Aires has allowed for a trustworthy record of detailed information regarding dengue cases (e.g., date of onset of symptoms, exact location) that are often not available. Studying the relationship between symptoms of both affections and laboratory diagnosis confirmation would help to optimize future records of both dengue and COVID-19, enabling proper diagnosis and treatment.

In the world, the effect of compulsory social isolation (lockdown) due to COVID-19 pandemic on dengue transmission has been variable, with some countries recording more cases than usual and others the opposite (Brady & Wilder-Smith, 2021). The dengue epidemic of 2020 was the strongest in Argentina since the reemergence of the virus in 1998, and mobility restriction measures due to COVID-19 coincidence did not contain its transmission. The crisis unleashed by the pandemic once again highlights the unequal environmental, health and economic conditions in which part of our society lives. As long as the lack of social protection and precarious conditions in which the underprivileged sectors of society live continues, as well as the fragility of the health systems to which they have access, diseases associated with overcrowding, lack of infrastructure, lack of safe water, disorganized urbanization, among other conditions that favor the development of infections, will continue to exist. Although 2020 will be worldwide remembered for the onset of COVID-19 pandemics, health care towards other needs and diseases should not be unattended, especially in territories with pre-existing vulnerabilities.

This work was possible thanks to the health system workers who carried out the compilation and notification to the National Surveillance System of COVID-19 and Dengue cases throughout the Buenos Aires City territory. We extend our gratitude to the National and Buenos Aires City Epidemiology Departments, and the clinical and laboratory surveillance coordinators for their role in coordinating the entire surveillance network in this jurisdiction. We are also grateful to the National Health Ministry authorities Carla Vizzotti, Analía Rearte and Carlos Giovacchini for their support. Finally, to Mariana Marcos for providing the shapefile layer on socio-economic categories and to Mariana Mauriño for her work on the dengue databases consolidation.

Additional Information and Declarations

Competing Interests

Author Contributions

Data Availability

The authors declare there are no competing interests.

Aníbal E. Carbajo conceived and designed the experiments, performed the experiments, analyzed the data, prepared figures and/or tables, authored or reviewed drafts of the article, and approved the final draft.

María V. Cardo conceived and designed the experiments, performed the experiments, analyzed the data, prepared figures and/or tables, authored or reviewed drafts of the article, and approved the final draft.

Martina Pesce conceived and designed the experiments, performed the experiments, analyzed the data, prepared figures and/or tables, authored or reviewed drafts of the article, and approved the final draft.

Luciana E. Iummato conceived and designed the experiments, performed the experiments, analyzed the data, prepared figures and/or tables, authored or reviewed drafts of the article, and approved the final draft.

Pilar Bárcena Barbeira conceived and designed the experiments, performed the experiments, analyzed the data, prepared figures and/or tables, authored or reviewed drafts of the article, and approved the final draft.

María Soledad Santini conceived and designed the experiments, authored or reviewed drafts of the article, supervised, and approved the final draft.

María Eugenia Utgés conceived and designed the experiments, performed the experiments, authored or reviewed drafts of the article, supervised, and approved the final draft.

The following information was supplied regarding data availability:

The dataset for the joint analysis of dengue and COVID-19 syndemic in Buenos Aires Authonomous City (Argentina), from epidemiological week 1 to 22, 2020 is available at figshare: Carbajo, Aníbal E.; Cardo, María V; Pesce, Martina; Iummato, Luciana E.; Barbeira, Pilar Barcena; Santini, María Soledad; et al. (2022): Data from the publication: “Age and socio-economic status affect dengue and COVID-19 incidence: Spatio-temporal analysis of the 2020 syndemic in Buenos Aires City”. figshare. Dataset. https://doi.org/10.6084/m9.figshare.21502416.v1.

Each row represents a census radius (the minimal census unit by INDEC).

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
