# Peer review of "Age and socio-economic status affect dengue and COVID-19 incidence: spatio-temporal analysis of the 2020 syndemic in Buenos Aires City"

_PeerJ, doi:10.7717/peerj.14735_

## Round 0.1 · original submission · Major Revisions

Specific comments for revisions (e.g. on Figures 1, 2 and 5) were given. Please address all of those comments.

Reviewer 1 ·

Basic reporting

Regarding Figure 2, it would be better to use the same notation for the legend (A-D) and the figure (a-d).

Regarding Figure 5, it is unclear see the difference between combined risks 1 and 2, 20 and 25. It would be more reader-friendly to change the color of the legend.

Experimental design

Are the temperatures depicted in Figure 2 the weekly average? It would be better to clarify this.

The data used in the study was obtained from the National Health Surveillance System. How good is this database at reflecting the actual infection situation? Are there any differences in ascertainment by area, and how much do they affect the results? If there is a difference, should this be considered?

Validity of the findings

The authors discussed the impact of the public health response that was implemented for COVID-19. In the introduction, the authors mentioned the measures that were implemented, but were these measures continued during the study period?

The "combined risk" was defined as the product of the respective categories of dengue and COVID-19 (COVID-19 category * dengue category). Did the authors define these? I understand that combined risk 1 and 2 are low risk and 20 and 25 are high risk, but what about the variance of each incidence?

Reviewer 2 ·

Basic reporting

The manuscript is very well written. With the methodological design very consistent with the objectives presented.
The introduction is very well constructed with many references and contextualization of the problem addressed in the manuscript.

The methods chosen are in line with the work proposal. However, I think it is necessary to present a study of the quality of both Dengue and COVID databases. It is important to know the percentage of duplicate records in the database as well as the percentage of completion of critical variables such as addressing and age, for example.

As important as the study of the quality of filling out the epidemiological databases is the information on the percentage of total cases that were georeferenced at each stage of the geocoding method used. Because we know the bias that the most vulnerable areas have in relation to the precariousness in the issue of addresses, which reflects their invisibility before the public power.

The results are consistent with the objectives and hypotheses raised. However, I believe that they can be better represented graphically to value them more and facilitate the reader's understanding and analysis. Figure 1 in particular needs to be revised as it is very confusing. It needs to have the divisions of neighborhoods, administrative regions. In addition to improving the grayscale to represent the scales of precariousness. It would be interesting if there were two figures: one with the issue of political-administrative limits (with a satellite image underneath allowing the observation of the urban fabric) and another with the classification and identification of favelas.

Discussion is adequate and converses with the literature on the topic.

Experimental design

The object of the manuscript is within the scope of the journal. It is extremely relevant for filling gaps in knowledge about this new disease that has affected all humanity.

Research that presented clear methods and that with few adjustments could be even better.

Validity of the findings

The results presented are robust and were well constructed, enabling the arrival of important conclusions on the subject.

---

## Round 0.2 · accepted · Accept

Your response has been quite successful.

Reviewer 1 ·

Basic reporting

no comment

Experimental design

no comment

Validity of the findings

no comment

Reviewer 2 ·

Basic reporting

no comment

Experimental design

no comment

Validity of the findings

no comment

Additional comments

no comment